# Isolation and Characterization of Antibacterial Carotane Sesquiterpenes from *Artemisia argyi* Associated Endophytic *Trichoderma virens* QA-8

**DOI:** 10.3390/antibiotics10020213

**Published:** 2021-02-20

**Authors:** Xiao-Shan Shi, Yin-Ping Song, Ling-Hong Meng, Sui-Qun Yang, Dun-Jia Wang, Xing-Wang Zhou, Nai-Yun Ji, Bin-Gui Wang, Xiao-Ming Li

**Affiliations:** 1Key Laboratory of Experimental Marine Biology, Institute of Oceanology, Chinese Academy of Sciences, and Laboratory of Marine Biology and Biotechnology, Qingdao National Laboratory for Marine Science and Technology, Nanhai Road 7, Qingdao 266071, China; shixs@qdio.ac.cn (X.-S.S.); menglh@ms.qdio.ac.cn (L.-H.M.); yangsuiqun@qdio.ac.cn (S.-Q.Y.); 2Yantai Institute of Coastal Zone Research, Chinese Academy of Sciences, Yantai 264003, China; ypsong@yic.ac.cn (Y.-P.S.); nyji@yic.ac.cn (N.-Y.J.); 3College of Chemistry and Chemical Engineering, Hubei Normal University, Cihu Road 11, Huangshi 435002, China; dunjiawang@hbnu.edu.cn (D.-J.W.); zhouxw@hbnu.edu.cn (X.-W.Z.); 4Center for Ocean Mega-Science, Chinese Academy of Sciences, Nanhai Road 7, Qingdao 266071, China

**Keywords:** *Artemisia argyi*, endophytic fungus, *Trichoderma virens*, carotane sesquiterpenes, antibacterial activity

## Abstract

Carotane sesquiterpenes are commonly found in plants but are infrequently reported in the fungal kingdom. Chemical investigation of *Trichoderma virens* QA-8, an endophytic fungus associated with the inner root tissue of the grown medicinal herb *Artemisia argyi* H. Lév. and Vaniot, resulted in the isolation and characterization of five new carotane sesquiterpenes trichocarotins I–M (**1**–**5**), which have diverse substitution patterns, and seven known related analogues (**6**–**12**). The structures of these compounds were established on the basis of a detailed interpretation of their NMR and mass spectroscopic data, and the structures including the relative and absolute configurations of compounds **1**–**3**, **5**, **9**, and **10** were confirmed by X-ray crystallographic analysis. In the antibacterial assays, all isolates exhibited potent activity against *Escherichia coli* EMBLC-1, with MIC values ranging from 0.5 to 32 µg/mL, while 7*β*-hydroxy CAF-603 (**7**) strongly inhibited *Micrococcus luteus* QDIO-3 (MIC = 0.5 µg/mL). Structure-activity relationships of these compounds were discussed. The results from this study demonstrate that the endophytic fungus *T. virens* QA-8 from the planted medicinal herb *A. argyi* is a rich source of antibacterial carotane sesquiterpenes, and some of them might be interesting for further study to be developed as novel antibacterial agents.

## 1. Introduction

*Artemisia argyi* H. Lév. and Vaniot, a traditional medicinal and edible plant in China, is reported to have pleiotropic bioactivities and has been used for the treatment of a variety of diseases, such as amenorrhea, bruising, dysmenorrhea, inflammation, jaundice, malaria, and metrorrhagia [1,2,3]. This plant has been widely cultivated as a medicinal herb in Qichun county, Hubei province, China. *Artemisia argyi* and its endophytic fungi (*Trichoderma koningiopsis* and *T. virens*) have been the source of a wide range of biologically active natural products [4,5,6,7,8,9]. A number of bioactive constituents such as flavonoids [4], polysaccharides [5], terpenes [6,7], and polyketides [8,9], have been reported from the plant *A. argyi* and its associated endophytic fungi. Several of these components have been investigated to show anticoagulation, antioxidant, antimicrobial, and anticancer activities [4,5,6,7,8,9,10].

Endophytic fungi have recently become an important source of new chemical substances with interesting biological activities. A wide range of compound classes such as alkaloids, quinones, phenols, polyketides, and terpenoids have been reported from various sourced endophytic fungi, and some of these compounds have been evidenced to possess significant biological properties such as antibacterial, anticancer, antifungal, antiinfections, antioxidant, antivirus, and enzymatic inhibitory activities [6,8,9,11,12,13]. Paclitaxel and penicillin, two famous chemicals that are well developed and marketed, could also be produced by endophytic fungi [12]. Based on the above results, we undertook a chemical investigation into the endophytic fungus from *A. argyi* which resulted in the identification of six cadinane-type sesquiterpenes trichocadinins B–G from *Trichoderma virens* QA-8, an endophytic fungus associated with the inner root tissue of *A. argyi* [6]. Further work on the additional portions of the culture extract led to the identification of 12 carotane sesquiterpenes (**1**–**12**) (Figure 1), with five new (trichocarotins I–M, **1**–**5**) and seven known (**6**–**12**) [14,15,16,17] related analogues. Previous studies have shown that carotane sesquiterpenes have pleiotropic bioactivities, such as antifungal activity against *Candida albicans* strains, antifertility, anti-HIV activity, and inhibitory activity on the growth of etiolated wheat coleoptiles [14,17,18,19,20,21]. This paper describes the isolation, structure determination, stereochemical assignment, and antibacterial activities of the isolated compounds, and the results indicates that *T. virens* QA-8, an endophytic fungus from the planted medicinal herb *A. argyi*, has abundant antibacterial carotene sesquiterpenoids.

## 2. Results and Discussion

### 2.1. Structural Elucidation of the New Compounds

Compound **1** was purified as colorless crystals and its molecular formula was determined as C_15_H_26_O_3_ on the basis of HRESIMS data, implying three degrees of unsaturation. The ^1^H and ^13^C NMR data of **1** indicated the presence of four methyls, three methylenes, five methines (including one olefinic and two oxygenated), and three non-protonated carbons (including one olefinic and one oxygenated) (Table 1 and Table 2 and Appendix A). Comprehensive analysis of its ^1^H and ^13^C NMR data suggested that compound **1** belonged to carotane sesquiterpenes with structural similarity to that of CAF-603 (**6**, Figure 1) and differed from **6** mainly at the five-membered ring [14,15]. Compared to CAF-603 (**6**), resonances for the methylene group at *δ*_H_/*δ*_C_ 1.39/50.3 (CH_2_-2) of **6** disappeared in that of **1**. Instead, signals corresponding to an oxymethine group at *δ*_H_/*δ*_C_ 3.09/78.4 (CH-2) were observed in the NMR spectra of **1**. Additionally, a signal for an additional hydroxy group was observed at *δ*_H_ 4.16 (s, OH-2) in the ^1^H NMR spectrum of **1**. These data suggested that the methylene group CH_2_-2 in **6** was replaced by an oxymethine group in **1**. COSY and HMBC data (Figure 2) supported the above deduction. The planar structure of **1** was thus determined.

The relative configuration of **1** was determined by the observed NOEs from H_3_-15 to the protons of OH-2 and OH-4, from H-10α to the proton of OH-2, and from H-5 to H-2, H-3, and H-10*β*. The Cu Kα radiation single-crystal X-ray diffraction experiment resulted in Flack parameter 0.1(4) of **1**, which allowed the establishment of its absolute configuration as 1*S*, 2*R*, 3*R*, 4*S*, and 5*S* (Figure 4). Thus, the structure of **1** was identified and named as trichocarotin I.

Trichocarotin J (**2**) was originally isolated as an amorphous powder. Its molecular formula was also determined as C_15_H_26_O_3_ based on the HRESIMS data. The ^1^H NMR spectrum of **2** indicated the presence of an olefinic methyl moiety, an aliphatic singlet methyl, and an isopropyl unit, which are typical structural features of the known carotane sesquiternene members. The ^1^H, ^13^C, and DEPT NMR data of **2** (Table 1 and Table 2 and Appendix A) showed almost identical spectral patterns to those of 7*β*-hydroxy CAF-603 (**7**) [15], with some minor variations for the chemical shifts of C-5 through C-9, and C-14 as well. Detailed inspection of the NMR data suggested that **2** is a diastereomer of **7**, epimeric at C-7. This was supported by the observed NOEs from H-5 to H-3, H-6*β,* H-7, and H-10*β*, and from H_3_-15 to the protons of OH-3 and OH-4 (Figure 3). Upon slow evaporation of the solvent (MeOH:H_2_O = 8:1) by storage in a refrigerator, quality single crystals of compound **2** were obtained. The structure and absolute configuration of **2** were further confirmed by a single-crystal X-ray diffraction experiment using Cu Kα radiation (Figure 4). The Flack parameter 0.06(4) allowed for the establishment of the absolute configuration of **2** as 1*R*, 3*R*, 4*S*, 5*S*, and 7*R*.

The molecular formula of trichocarotin K (**3**) was determined to be C_15_H_26_O_3_ by HRESIMS. Its ^1^H and ^13^C NMR data revealed the presence of four methyls, four methylenes, four methines (one oxygenated), and three non-protonated (one oxygenated and one ketone) carbons (Table 1 and Table 2 and Appendix A). A detailed comparison of NMR data revealed that compound **3** differed from CAF-603 (**6**) [14,15] mainly at the seven-membered ring, and that resonances corresponding to the double bond (C-8 and CH-9) in **6** disappeared in that of **3**, while signals for an aliphatic methine (CH-8) and for a keto group (C-9) were observed in the NMR spectra of **3** (Table 1 and Table 2). HMBC correlations from H-14 to C-7, C-8, and C-9 confirmed this deduction. Other COSY and HMBC correlations (Figure 2) further confirmed the planar structure of **3**.

The key NOE correlations from H_3_-15 to H-8, H-10α, and to the protons of OH-3 and OH-4, and from H-3 to H-5 determined the relative configuration of **1** (Figure 3). Upon slow evaporation of the solvent (MeOH) by storing the sample in a refrigerator, quality single crystals of **3** were obtained, and the absolute configuration of **3** was thus determined as 1*S*, 3*R*, 4*S*, 5*S*, and 8*S* by X-ray diffraction analysis (Figure 5).

Compound **4** was isolated as a colorless oil and its molecular formula was determined as C_15_H_24_O_2_ by HRESIMS. Its ^1^H and ^13^C NMR data revealed the presence of four methyls, two methylenes, six methines (one oxygenated and three olefinic), and three non-protonated (one oxygenated and one olefinic) carbons (Table 1 and Table 2 and Appendix A). These data suggested that **4** had the same carbon skeleton as that of CAF-603 (**6**) [14,15]. Actually, compound **4** was a C-6 and C-7 deprotonated product of CAF-603. This was verified by chemical shifts and COSY correlations of H-6 with H-5 and H-7 (Figure 2). In the NOESY experiments, the NOE from H-3 to H-5 indicated the co-facial orientation of these groups (Figure 3). However, no other diagnostic NOEs were observed, and thus the relative configuration could not be characterized by NOESY experiments. According to the literature reports [14,15,16], the relative configuration of all the congeners of CAF-603 was deduced to be same at the five-membered ring. The absolute configuration of **4** was tentatively deduced as 1*R*, 3*R*, 4*S*, 5*S* on the basis of biogenic considerations. Thus, the structure of compound **4** was characterized and was named trichocarotin L.

Trichocarotin M (**5**), initially obtained as a colorless waxy solid, was determined to possess the molecular formula C_15_H_26_O_3_, based on HRESIMS data. The ^1^H and ^13^C NMR data of **5** (Table 1 and Table 2 and Appendix A) closely resembled those of 14-hydroxy CAF-603 (trichocarane B), whose relative configuration was determined by NOESY spectrum [17], but its absolute configuration was not clarified. The relative configuration of **5** was deduced to be the same as that of 14-hydroxy CAF-603 based on the NOESY data (Figure 3). However, the optical rotation of compound **5** ([α]25 D +16 (*c* 0.25, CHCl_3_)) has an opposite sign to that of 14-hydroxy CAF-603 ([α]25 D –28.0 (*c* 0.20, CHCl_3_)) [17]. Thus, trichocarotin J (**5**) might be the enantiomer of 14-hydroxy CAF-603. To further confirm this deduction we turned our efforts to a crystallographic study of this compound. Upon slow evaporation of the solvent (MeOH:H_2_O = 15:1) by storage in a refrigerator, quality single crystals of compound **5** were obtained, and the relative and absolute configuration of **5** was therefore established by a single-crystal X-ray diffraction experiment using Cu Kα radiation (Figure 5). The Flack parameter 0.3(2) of **5** allowed the unambiguous confirmation of the absolute configuration of **5** as 1*R*, 3*R*, 4*S*, and 5*S*, and named trichocarotin M, which is an enantiomer of 14-hydroxy CAF-603.

In addition to new compounds **1**–**5**, seven known analogues including CAF-603 (**6**) [14,15], 7*β*-hydroxy CAF-603 (**7**) [15], trichocarotins E–H (**8**–**11**) [16], and trichocarane A (**12**) [17] were also isolated and identified. Their structures were elucidated by comparing their NMR data with those reported in the literature. It deserve to mention that the absolute configurations of compounds **9** and **10** were previously assigned solely from a biogenetic perspective [16], but in the present study the single-crystal X-ray diffraction (Figure 6) was used to confirm their absolute configurations.

### 2.2. Antibacterial Activities of the Isolated Compounds

Compounds were assayed for their antibacterial activities against human pathogens *Escherichia coli* EMBLC-1 and *Micrococcus luteus* QDIO-3. Statistical analysis by ANOVA for various concentrations when assessed for *E. coli* and *M. luteus* showed potent inhibitory growth both at concentrations ≥ MICs when compared to that at concentrations < MICs, and the MICs were obtained. Significance of all the statistical tests was predetermined at *p* < 0.05. As a result, each of these compounds showed strong inhibitory activity against *E. coli*, with MIC values ranging from 0.5 to 16 µg/mL, and the activity of compounds **3**–**5**, **8**, and **11** are as active as that of the positive control (chloramphenicol, MIC = 0.5 µg/mL) (Table 3). In addition, compounds **6**–**8** showed potent activity against *M. luteus* with MIC values of 4, 0.5, and 2 µg/mL, respectively, and the activity of compound **7** was stronger than that of chloramphenicol (MIC = 1 µg/mL).

Structure-activity relationship analysis revealed that the OH substitution at C-11 increased the activities against both *E. coli* and *M. luteus* (**8** vs. **6**), while the OH group at C-14 increased the activity against *E. coli* and decreased the activity against *M. luteus* (**5** vs. **6**). Similarly, the 8,9-epoxy group at the seven-membered ring also increased the activity against *E. coli* and decreased the activity against *M. luteus* (**12** vs. **6**).

## 3. Materials and Methods

### 3.1. General Experimental Procedures

The general experimental procedures and the apparatus used in the current work are the same as that described in our previous report [6]: an SGW X-4 micro-melting-point apparatus, an Optical Activity AA-55 polarimeter, a PuXi TU-1810 UV-visible spectrophotometer, a JASCO J-715 spectropolarimeter, a Bruker Avance 500 spectrometer, an API QSTAR Pulsar 1 mass spectrometer, analytical HPLC: a Dionex HPLC system, equipped with P680 pump (Dionex), silica gel GF254 precoated plates, 100–200 mesh and 200–300 mesh silica gel, 40–63 μm RP-18 reverse-phase Si gel, and Sephadex LH-20.

### 3.2. Plant and Fungal Materials

The fungus *T. virens* QA-8 was isolated from the fresh inner root tissue of the Compositae medical plants *A. argyi* collected at Qichun, Hubei Province, in central China in July 2014 and was identified by analysis of its ITS region of the rDNA. The primers used for PCR are ITS1 (5′-TCCGTAGGTGAACCTGCGG-3′) and ITS4 (5′-TCCTCCGCTTATTGATATGC-3′), and the total length of sequenced ITS is 616bp. The BLAST search showed that the amplified ITS sequence (GenBank accession no. MK224593) has 100% homology with other members of the genus *T. virens* (compared with MT256290.1). The strain QA-8 is cryopreserved at −80 °C in 20% aqueous glycerol at the Key Laboratory of Experimental Marine Biology, Institute of Oceanology of the Chinese Academy of Sciences (IOCAS).

### 3.3. Fermentation, Extraction and Isolation

The culture of *T. virens* QA-8 was grown on PDA medium at 28 °C for 7 days and were then inoculated in a 1L-Erlenmeyer flask containing solid rice medium consisting of 70 g rice, 0.1 g corn flour, 0.3 g peptone, and 100 mL distilled water. After 30 days, the whole fermented cultures (180 flasks) were extracted with EtOAc and the combined EtOAc solution was concentrated under reduced pressure to yield 97.4g extract. The extract was fractionated by Si gel vacuum liquid chromatography (VLC) to yield 10 fractions (Frs. 1–10). Purification of Fr. 3 (3.0 g), Fr. 4 (3.7 g), Fr. 5 (11.0 g), and Fr. 6 (22.7 g) by reversed-phase column chromatography (CC) over Lobar LiChroprep RP-18 with a MeOH-H_2_O gradient (from 10:90 to 100:0) yielded six subfractions (Frs. 3.1–3.6), nine subfractions (Frs. 4.1–4.9), 10 subfractions (Frs. 5.1–5.10), and 15 subfractions (Frs. 6.1–6.15), respectively. Fr. 3.5 (341 mg) was further purified by CC on Si gel eluting with a PE–acetone gradient (from 50:1 to 10:1) to obtain compound **6** (185.9 mg). Fr. 4.9 (325 mg) was purified by CC on Si gel eluting with a CH_2_Cl_2_–acetone gradient (from 20:1 to 5:1) and then by preparative TLC (plate: 20 × 20 cm, developing solvents: PE–EtOAc, 1:1) to yield compounds **3** (7.9 mg) and **10** (18.3 mg). Fr. 5.5 (436 mg) was purified by CC on Si gel eluting with a CH_2_Cl_2_−acetone gradient (from 100:1 to 20:1) and then purified by CC on Sephadex LH-20 (MeOH) to give compounds **7** (3.6 mg), **8** (8.9 mg) and **12** (9.3 mg), respectively. Fr. 5.8 (275 mg) was purified by CC on Si gel eluting with a CH_2_Cl_2_−acetone gradient (from 100:1 to 10:1) and then purified by CC on Sephadex LH-20 (MeOH) and preparative TLC (plate: 20 × 20 cm, developing solvents: PE/EtOAc, 1:1) yielded compounds **2** (7.2 mg) and **9** (5.5 mg). Fr. 6.2 (136 mg) was further purified by preparative TLC (plate: 20 × 20 cm, developing solvents: CH_2_Cl_2_− EtOAc, 5:1) to afford compound **1** (6.5 mg). Fr. 6.7 (94 mg) was purified by CC on Sephadex LH-20 (MeOH) to give compound **11** (5.8 mg). Fr. 6.8 (582 mg) was purified by CC on Si gel eluting with a CH_2_Cl_2_−MeOH gradient (from 200:1 to 10:1) and then purified by preparative TLC (plate: 20 × 20 cm, developing solvents: PE/acetone/ acetic acid, 3:1:0.04) to obtain compounds **4** (8.9 mg) and **5** (12.0 mg).

### 3.4. Spectroscopic Data

*Trichocarotin I* (**1**): colorless crystals; mp 99–101 °C; [α]25 D –5.56 (*c* 0.18, MeOH); ECD (3.15 mM, MeOH) *λ*_max_ (Δ*ε*) 215 (–3.35), 249 (+1.75) nm; ^1^H and ^13^C NMR data, Table 1 and Table 2; HRESIMS *m/z* 253.1812 [M – H]^–^ (calcd for C_15_H_25_O_3_, 253.1809).

*Trichocarotin J* (**2**): colorless crystals; mp 88–90 °C; [α]25 D –10.64 (*c* 0.47, MeOH); ECD (4.33 mM, MeOH) *λ*_max_ (Δ*ε*) 223 (–28.47), 262 (+20.03) nm; ^1^H and ^13^C NMR data, Table 1 and Table 2; HRESIMS *m/z* 253.1810 [M – H]^–^ (calcd for C_15_H_25_O_3_, 253.1809).

*Trichocarotin K* (**3**): colorless crystals; mp 92–93 °C; [α]25 D +98.41 (*c* 0.63, MeOH); ECD (5.12 mM, MeOH) *λ*_max_ (Δ*ε*) 219 (+5.15), 234 (+5.39), 294 (+7.79) nm; ^1^H and ^13^C NMR data, Table 1 and Table 2; HRESIMS *m/z* 253.1808 [M – H]^–^ (calcd for C_15_H_25_O_3_, 253.1809).

*Trichocarotin L* (**4**): colorless oil; [α]25 D +26.32 (*c* 0.19, MeOH); ECD (7.20 mM, MeOH) *λ*_max_ (Δ*ε*) 243 (–10.60) nm; ^1^H and ^13^C NMR data, Table 1 and Table 2; HRESIMS *m/z* 235.1709 [M – H]^–^ (calcd for C_15_H_23_O_2_, 235.1704).

*Trichocarotin M* (**5**): colorless crystals; mp 131–133 °C; [α]25 D +85.92 (*c* 0.71, MeOH) or [α]25 D +16.00 (*c* 0.25, CHCl_3_); ECD (5.51 mM, MeOH) *λ*_max_ (Δ*ε*) 209 (–4.84), 220 (+2.44), 240 (–1.72) nm; ^1^H and ^13^C NMR data, Table 1 and Table 2; HRESIMS *m/z* 299.1869 [M + HCOO]^–^ (calcd for C_16_H_27_O_5_, 299.1864).

### 3.5. X-Ray Crystallographic Analysis of Compounds 1–3, 5, 9, and 10

The crystallographic data were collected on a Bruker Smart1000 or Bruker D8 Venture CCD diffractometer. The data were corrected for absorption by using the program SADABS [22,23]. The structures were solved by direct methods with the SHELXTL software package [24]. The structures were refined by full-matrix least-squares techniques [25].

*Crystal data for compound***1**: C_15_H_26_O_3_, F.W. = 254.36, Monoclinic space group *P*2(1), unit cell dimensions *a* = 6.9632(6) Å, *b* = 12.8384(13) Å, *c* = 16.8664(18) Å, *V* = 1506.5(3) Å^3^, α = *γ* = 90°, *β* = 92.3770(10)°, *Z* = 4, *d*_calcd_ = 1.121 mg/m^3^, crystal dimensions 0.36 × 0.08 × 0.05 mm^3^, µ = 0.605 mm^–1^, *F*(000) = 560. The 3967 measurements yielded 2231 independent reflections. The final refinement gave *R*_1_ = 0.0649 and w*R*_2_ = 0.1385 [*I* > 2*σ*(*I*)]. The Flack parameter was 0.1(4).

*Crystal data for compound***2**: C_15_H_26_O_3_·H_2_O, F.W. = 272.37, Orthorhombic space group *P*2(1)2(1)2(1), unit cell dimensions *a* = 7.7040(9) Å, *b* = 13.802(4) Å, *c* = 14.5355(16) Å, *V* = 1545.6(5) Å^3^, α = *β* = *γ* = 90°, *Z* = 4, *d*_calcd_ = 1.171 mg/m^3^, crystal dimensions 0.18 × 0.16 × 0.12 mm^3^, µ = 0.668 mm^–1^, *F*(000) = 600. The 2796 measurements yielded 2746 independent reflections. The final refinement gave *R*_1_ = 0.0293 and w*R*_2_ = 0.0804 [*I* > 2*σ*(*I*)]. The Flack parameter was 0.06(4).

*Crystal data for compound***3**: C_15_H_26_O_3_, F.W. = 254.36, Orthorhombic space group *P*2(1)2(1)2(1), unit cell dimensions *a* = 7.4144(10) Å, *b* = 7.5400(10) Å, *c* = 26.394(4) Å, *V* = 1475.5(4) Å^3^, α = *β* = *γ* = 90°, *Z* = 4, *d*_calcd_ = 1.145 mg/m^3^, crystal size 0.22 × 0.20 × 0.18 mm, µ = 0.617 mm^–1^, *F*(000) = 560. The 2675 measurements yielded 2588 independent reflections after. The final refinement gave *R*_1_ = 0.0347 and w*R*_2_ = 0.1193 [*I* > 2*σ*(*I*)]. The Flack parameter was −0.1(3).

*Crystal data for compound***5**: C_15_H_26_O_3_, F.W. = 254.36, monoclinic space group *P*2(1), unit cell dimensions *a* = 7.6721(12) Å, *b* = 7.4046(12) Å, *c* = 13.155(3) Å, *V* = 746.3(3) Å^3^, α = *γ* = 90°, *β* = 92.961(14)°, *Z* = 2, *d*_calcd_ = 1.132 mg/m^3^, crystal dimensions 0.16 × 0.15 × 0.14 mm, µ = 0.610 mm^–1^, *F*(000) = 280. The 2201 measurements yielded 1459 independent reflections. The final refinement gave *R*_1_ = 0.0542 and w*R*_2_ = 0.1530 [*I* > 2*σ*(*I*)]. The Flack parameter was 0.3(2) in the final refinement for all 2201 reflections with 1459 Friedel pairs.

*Crystal data for compound***9**: C_15_H_26_O_3_, F.W. = 254.36, Orthorhombic space group *P*2(1)2(1)2(1), unit cell dimensions *a* = 6.9592(2) Å, *b* = 11.0588(4) Å, *c* = 19.4762(9) Å, *V* = 1498.90(10) Å^3^, α = *β* = *γ* = 90°, *Z* = 4, *d*_calcd_ = 1.127 mg/m^3^, crystal dimensions 0.30 × 0.10 × 0.07 mm^3^, µ = 0.608 mm^–1^, *F*(000) = 560. The 2603 measurements yielded 2124 independent reflections. The final refinement gave *R*_1_ = 0.0403 and w*R*_2_ = 0.0908 [*I* > 2*σ*(*I*)]. The Flack parameter was 0.1(3).

*Crystal data for compound***10**: 2C_15_H_24_O_4_·CH_3_OH, F.W. = 568.73, Orthorhombic space group *P*2(1)2(1)2(1), unit cell dimensions *a* = 7.2324(7) Å, *b* = 15.2789(15) Å, *c* = 28.581(3) Å, *V* = 3158.3(5) Å^3^, α = *β* = *γ* = 90°, *Z* = 4, *d*_calcd_ = 1.196 mg/m^3^, crystal size 0.21 × 0.08 × 0.03 mm, µ = 0.704 mm^–1^, *F*(000) = 1240. The 5499 measurements yielded 1880 independent reflections. The final refinement gave *R*_1_ = 0.1114 and w*R*_2_ = 0.2422 [*I* > 2*σ*(*I*)]. The Flack parameter was 0.0(7) in the final refinement for all 5499 reflections with 1880 Friedel pairs.

### 3.6. Antibacterial Assays

Antibacterial evaluation against human pathogens *E. coli* EMBLC-1 and *M. luteus* QDIO-3 was carried out by the microplate assay with three repetitions [26]. *E. coli* and *M. luteus* (95 μL × 5 × 10^5^ CFU/mL per well) were cultured at 37 °C in LB medium containing 1% peptone, 0.5% yeast extract, 1% NaCl and distilled water with 5 μL various concentrations of compounds **1**–**12** in each well of 96-well plates for 24 h. The pathogens used in the assays were obtained from the Institute of Oceanology, Chinese Academy of Sciences. Chloramphenicol was used as positive control.

## 4. Conclusions

In summary, five new carotane sesquiterpenes trichocarotins I–M (**1**–**5**) and seven known analogues (**6**–**12**) were identified from the culture extract of endophytic fungus *Trichoderma virens* QA-8. Their structures were elucidated by a detailed interpretation of the spectroscopic data and the structures and absolute configurations of compounds **1**–**3**, **5**, **9**, and **10** were confirmed by X-ray crystallographic analysis. The crystal structures of the known compounds **9** and **10** are reported for the first time. The absolute configurations of this kind of sesquiterpene were barely presumed by biosynthesis in previously published reports [16], and some of them were not even determined [14,17], but in the present study single crystal X-ray diffraction was used to confirm their absolute configurations. Compounds **3**–**5**, **8**, and **11** showed inhibitory activity against *E. coli* (MIC = 0.5 µg/mL) and compound **7** showed the strongest activity against *M. luteus* (MIC = 0.5 µg/mL), which are similar to or stronger than that of the positive control. Our results suggested that some of these compounds could be interesting and could lead compounds into a further development of novel antibacterial agents.

## Figures and Tables

**Figure 1 antibiotics-10-00213-f001:**
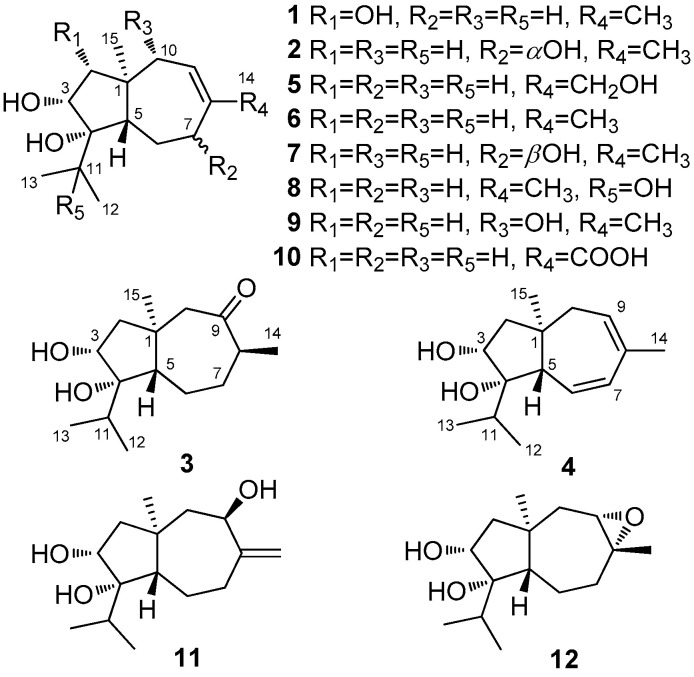
Structures of compounds **1**−**12**.

**Figure 2 antibiotics-10-00213-f002:**
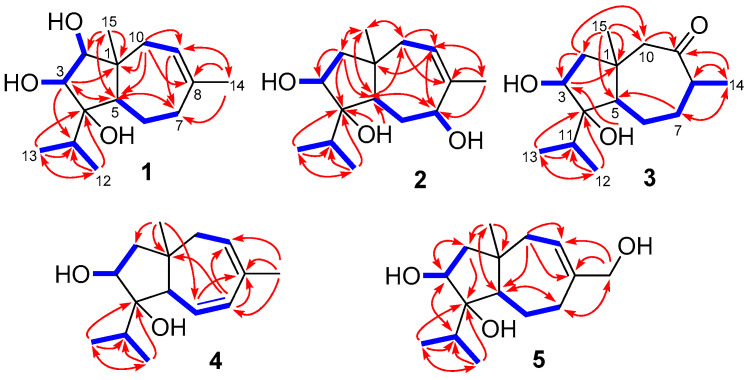
Key HMBC (arrows) and COSY (bold lines) correlations of compounds **1**−**5**.

**Figure 3 antibiotics-10-00213-f003:**
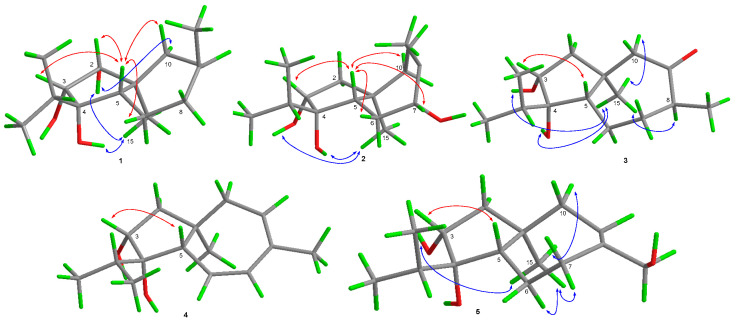
Key NOE correlations of compounds **1**–**5**.

**Figure 4 antibiotics-10-00213-f004:**
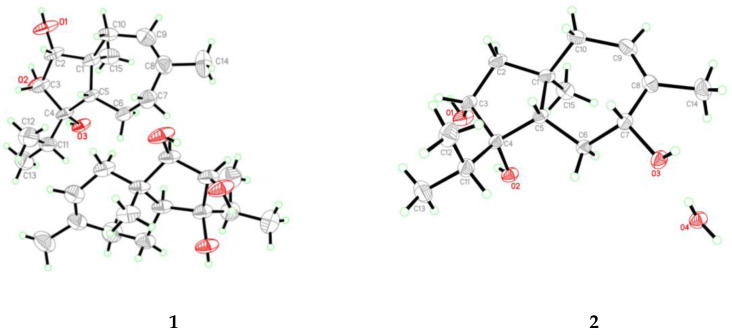
X-ray crystallographic structures of compounds **1** and **2**.

**Figure 5 antibiotics-10-00213-f005:**
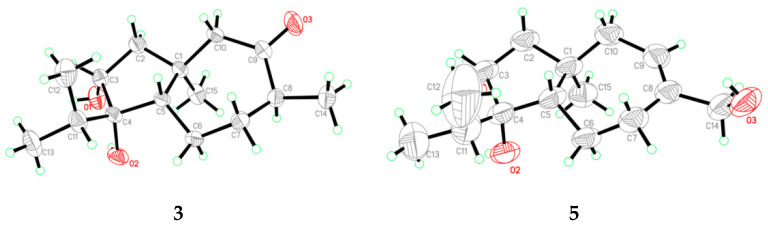
X-ray crystallographic structures of compounds **3** and **5**.

**Figure 6 antibiotics-10-00213-f006:**
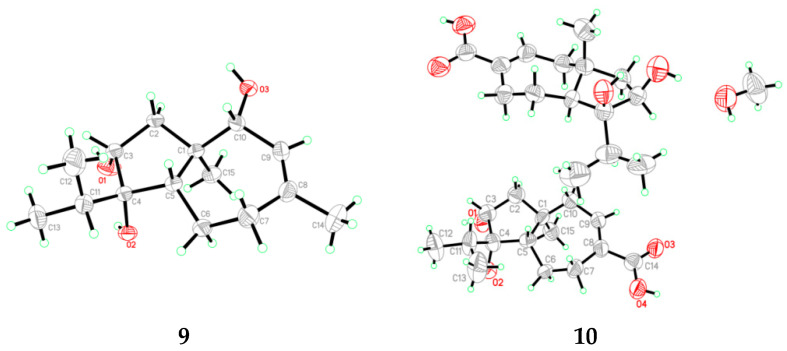
X-ray crystallographic structures of compounds **9** and **10**.

**Table 1 antibiotics-10-00213-t001:** ^1^H NMR Data for Compounds **1**–**5** in DMSO-*d*_6_ (500 MHz).

No.	1	2	3	4	5
2	3.09, d (7.0)	1.41, m (overlap)	1.50, m	1.64, m (overlap)	1.49, m (overlap)
3	3.63, d (7.0)	3.88, m	3.88, dd (7.2, 1.6)	3.87, m	3.88, t (4.4)
5	1.26, dd (11.4, 1.7)	1.41, m (overlap)	1.66, m	1.64, m (overlap)	1.36, m
6α	1.49, m	1.56, m	1.78, m	5.80, d (11.2)	1.30, m
6*β*	1.37, m	1.49, m	1.78, m		1.49, m (overlap)
7α	2.00, m (overlap)	4.03, m	1.43, m	5.39, d (11.2)	1.81, m (overlap)
7*β*	2.00, m (overlap)		1.43, m		2.11, dd (14.9, 3.2)
8			2.38, m		
9	5.32, br d (7.0)	5.26, m		5.57, m	5.50, br d (8.2)
10α	1.59, dd (14.4, 1.2)	1.87, dd (14.0, 9.1)	2.43, d (14.4)	2.18, m	1.81, m (overlap)
10*β*	2.07, dd (14.4, 8.9)	1.68, m	2.30, d (14.4)	2.18, m	1.99, dd (14.4, 8.9)
11	1.66, m	1.74, m	1.70, m	1.64, m (overlap)	1.69, m
12	0.79, d (6.9)	0.81, d (6.9)	0.79, d (6.9)	0.85, d (6.9)	0.81, d (6.9)
13	0.85, d (6.7)	0.88, d (6.9)	0.86, d (6.7)	0.78, d (6.9)	0.87, d (6.7)
14	1.70, s	1.71, s	0.96, d (6.9)	1.74, s	3.76, s
15	0.71, s	0.88, s	1.06, s	0.98, s	0.92, s
2-OH	4.16, s				
3-OH	5.39, s	5.21, d (4.2)	5.35, s		
4-OH	3.53, s	3.69, s	3.76, s		3.60, br s
7-OH		4.66, d (5.1)			

**Table 2 antibiotics-10-00213-t002:** ^13^C NMR Data for Compounds **1**–**5** in DMSO-*d*_6_ (125 MHz).

No.	1	2	3	4	5
1	44.5, C	40.7, C	40.2, C	44.6, C	41.6, C
2	78.4, CH	49.6, CH_2_	50.3, CH_2_	47.9, CH_2_	50.0, CH_2_
3	70.6, CH	70.1, CH	70.4, CH	71.8, CH	70.4, CH
4	79.0, C	82.5, C	82.1, C	82.1, C	82.6, C
5	54.3, CH	53.5, CH	55.4, CH	47.2, CH	57.7, CH
6	20.2, CH_2_	31.7, CH_2_	18.4, CH_2_	143.8, CH	21.1, CH_2_
7	34.1, CH_2_	72.6, CH	33.0, CH_2_	125.3, CH	30.0, CH_2_
8	138.4, C	143.6, C	46.3, CH	129.0, C	142.6, C
9	122.3, CH	119.6, CH	214.3, C	127.4, CH	121.4, CH
10	40.0, CH_2_	41.5, CH_2_	57.3, CH_2_	27.0, CH_2_	42.1, CH_2_
11	34.8, CH	34.4, CH	34.7, CH	35.0, CH	34.7, CH
12	17.6, CH_3_	17.7, CH_3_	17.6, CH_3_	17.7, CH_3_	17.7, CH_3_
13	17.1, CH_3_	17.2, CH_3_	17.1, CH_3_	17.3, CH_3_	17.1, CH_3_
14	27.1, CH_3_	21.1, CH_3_	22.8, CH_3_	26.9, CH_3_	66.9, CH_2_
15	13.9, CH_3_	20.9, CH_3_	23.2, CH_3_	21.4, CH_3_	20.9, CH_3_

**Table 3 antibiotics-10-00213-t003:** Antibacterial activity of compounds **1–12** (MIC, μg/mL).

No.	1	2	3	4	5	6	7	8	9	10	11	12	Chloramphenicol
EC	16	32	0.5	0.5	0.5	16	16	0.5	16	16	0.5	8	0.5
ML	–	32	–	–	8	4	0.5	2	–	–	32	8	1

EC: *E. coli*. ML: *M. luteus*; –: no activity (MIC > 64 µg/mL).

## Data Availability

The data presented in this study is included in the Appendix A and is available online at http://www.mdpi.com/xxx/s1.

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
