# Peer review of "Isolation and Characterization of Antibacterial Carotane Sesquiterpenes from Artemisia argyi Associated Endophytic Trichoderma virens QA-8"

_antibiotics, 2021, doi:10.3390/antibiotics10020213_

Round 1
Reviewer 1 Report
This paper describes the isolation, structure determination, stereochemical assignment, and antimicrobial activities of the isolated compounds from Artemisia argyi. The manuscript fits within the scope of the journal. The manuscript is interesting and well organized. The title is clear and it is adequate to the content of the article. The revisions are necessary to improve the clarity of the presentation.
I have some recommendations for authors:
- Introduction should be completed with recent literature data about the biological activity of endophytic fungus. In this sense, the use of relevant MS published in the last years and focused on antibacterial compounds could also be of great interest. Currently, further valuable information to the readers is needed, in order to offer a whole vision of the issue.
- please include paternity for the binomial name of a plant: Artemisia argyi H.Lév. & Vaniot
- please include brief information about antibacterial assay, even if it is the citation index.
- a problem is the lack of statistical analyses of the data. Analyses were conducted in triplicate? Without this informations any conclusions are superfluous. So, the statistical analysis of results may also be improved upon.
- compare results for antibacterial activities of the isolated compounds with other compounds from recent literature data.
- Include in the text potential research directions.
The author’s work on discussing achieved results is appreciated. The revisions are necessary to improve the clarity of the presentation and needed to make convincing scientific arguments.
Author Response
Response to Reviewer #1:
1, Introduction should be completed with recent literature data about the biological activity of endophytic fungus. In this sense, the use of relevant MS published in the last years and focused on antibacterial compounds could also be of great interest. Currently, further valuable information to the readers is needed, in order to offer a whole vision of the issue.
Response: Thanks to the suggestion. Some new references were cited accordingly (ref. no. 3, 12 and 13).
2, please include paternity for the binomial name of a plant: Artemisia argyi H.Lév. & Vaniot
Response: Corrected, as suggested, but not including the title.
3, please include brief information about antibacterial assay, even if it is the citation index.
Response: We have added the brief information about antibacterial assay.
4, a problem is the lack of statistical analyses of the data. Analyses were conducted in triplicate? Without this informations any conclusions are superfluous. So, the statistical analysis of results may also be improved upon.
Response: Thanks to the suggestion. Analyses were conducted in triplicate and we have added the brief information about antibacterial assay.
5, compare results for antibacterial activities of the isolated compounds with other compounds from recent literature data.
Response: Thanks to the suggestion. The bological activity of carotane sesquiterpenes was discussed. Some new references were cited accordingly (ref. no. 18-21). In previous reports (ref. no. 16), the known compounds have did been subjected to the antibacterial assays, but the tested strains are different from our work, and therefore cannot be compared.
6, Include in the text potential research directions.
Response: Thanks to the suggestion. Some more information was added in the “Conclusions”.
Reviewer 2 Report
Isolation and Charaterization of Antibaterial Carotane Sequiterpenes from Artemisia argyi Associated Endophytic Trichoderma virens QA-8.
The authors characterize several new carotane sesquiterpenes and related analoques. In addition, there is a small section on the antimicrobial activity of these compounds.
I found the article interesting and the article for the most part is well written and organized, with exception of the Introduction and one question about how many bacterial strains were assayed.
Main issues:
1) The Introduction currently consists of one paragraph of which it should be expanded for a better review of the past research. I would suggest suggestion 1) introducing the medicinal plant and its importance and relating it to the fungus, 2) brief background about what is and what is known about carotane sesquiterpenes, and finally 3) description of the purpose of this study.
2) Table 3 at the end shows seven bacterial species but the manuscript only states 2. If all bacterial species were used, then they need to be included in the methods and material and results and discussion.
Individual line items:
Abstract:
First sentence: .....but infrequently reported from fungal kingdom.Consider changing to: .... but infrequently reported from the fungal kingdom.
Last sentence: The results from this study disclosed that the endophytic fungus.....Suggest changing to: The results from this study demonstrate that the endophytic fungus....
Last part of the last sentence: with some of them might be interesting for further study to be developed as novel antibacterial agents.
Consider revising: I would stop the sentence before with, Start a new sentence and add. Our results suggested that some of these compounds could be interesting lead compounds for further development of novel antibacterial agents.
Introduction:
Second sentence: herb in Qichun county of Hubei province in China. A. argyi and its..Consider revising: herb in Qichun county, Hubei province, in China. Artemisia argyi and its..Genus is spelled out when it's the first word of a sentence.
Results and Discussion:
page 2: Paragraph below Figure 4: Trichocarotin J (2) was originally isolated as amorphous powder.Consider revising: Trichocarotin J (2) was originally isolated as an amorphous powder.
page 4: Section 2.2 and Table 3. Please look at this section. Underneath Table two there is a list of more than two bacteria that was tested. Only two E.coli and M. luteus are listed throughout the manuscript. If the others were tested and there was no inhibition, the authors still need to state this within the manuscript. Please review this and add the information throughout or if an older table was used and this is in error, please remove the additional bacteria.
last sentence in the first paragraph below 2.2: and the compound 7 is stronger than that of...Consider revising: and the compound 7 was stronger than that of...
Materials and Methods:
3.2 Plant and fungal materials
Fist sentence: collected at Qichun of Hubei Province in central China......Consider revising: collect at Quchun, Hubei Province, in central China....
3.3 Fermentation, extraction and isolation
First sentence: The fresh mycelia
Consider revising: Fresh mycelia
3.6 Antibacterial assays
There are only two bacterial species here but Table 3 at the end shows more. Please clarify as per the previous above comment.
Author Response
1) The Introduction currently consists of one paragraph of which it should be expanded for a better review of the past research. I would suggest suggestion 1) introducing, 2) brief background about what is and what is known about carotane sesquiterpenes, and finally 3) description of the purpose of this study.
Response: Thanks to the suggestion. The importance of the medicinal plant and relating it to the fungus were provided. Some new references were cited accordingly (ref. no. 3, 10 and 18-21).
2) Table 3 at the end shows seven bacterial species but the manuscript only states 2. If all bacterial species were used, then they need to be included in the methods and material and results and discussion.
Response: Checked and corrected, as suggested.
Individual line items:
Abstract:
First sentence: .....but infrequently reported from fungal kingdom. Consider changing to: .... but infrequently reported from the fungal kingdom.
Response: Checked and corrected, as suggested.
Last sentence: The results from this study disclosed that the endophytic fungus.....Suggest changing to: The results from this study demonstrate that the endophytic fungus....
Response: Checked and corrected, as suggested.
Last part of the last sentence: with some of them might be interesting for further study to be developed as novel antibacterial agents.
Consider revising: I would stop the sentence before with, Start a new sentence and add. Our results suggested that some of these compounds could be interesting lead compounds for further development of novel antibacterial agents.
Response: Checked and corrected, as suggested.
Introduction:
Second sentence: herb in Qichun county of Hubei province in China. A. argyi and its..Consider revising: herb in Qichun county, Hubei province, in China. Artemisia argyi and its..Genus is spelled out when it's the first word of a sentence.
Response: Checked and corrected, as suggested.
Results and Discussion:
page 2: Paragraph below Figure 4: Trichocarotin J (2) was originally isolated as amorphous powder.Consider revising: Trichocarotin J (2) was originally isolated as an amorphous powder.
Response: Checked and corrected, as suggested.
page 4: Section 2.2 and Table 3. Please look at this section. Underneath Table two there is a list of more than two bacteria that was tested. Only two E.coli and M. luteus are listed throughout the manuscript. If the others were tested and there was no inhibition, the authors still need to state this within the manuscript. Please review this and add the information throughout or if an older table was used and this is in error, please remove the additional bacteria.
Response: Checked and we have removed the additional bacteria.
last sentence in the first paragraph below 2.2: and the compound 7 is stronger than that of...Consider revising: and the compound 7 was stronger than that of...
Response: Checked and corrected.
Materials and Methods:
3.2 Plant and fungal materials
Fist sentence: collected at Qichun of Hubei Province in central China......Consider revising: collect at Quchun, Hubei Province, in central China....
Response: Checked and corrected.
3.3 Fermentation, extraction and isolation
First sentence: The fresh mycelia
Consider revising: Fresh mycelia
Response: Checked and we have changed to: The culture of T. virens QA-8 was grown on PDA medium at 28 °C for 7 days.
3.6 Antibacterial assays
There are only two bacterial species here but Table 3 at the end shows more. Please clarify as per the previous above comment.
Response: Checked and corrected.
Reviewer 3 Report
the manuscript is well-written, but still need some corrections. Some are suggested below:
Page 1. Abstract and through the text
the planted medicinal herb – the grown (cultivated) medicinal herbs
Introduction
widely cultured as an medicinal herb - widely cultivated as an medicinal herb
A.argyi and its endophytic fungus have been the source of a wide range of biologically active natural products. – Please, specify “its endophytic fungus” by name, and give reference, or remove this sentence.
Page 2.
Further work on the additional portions of the culture extract leading to the identification of 12 carotane sesquiterpenes (1–12) (Figure 1), with five new (trichocarotins I–M, 1–5) and seven known (6–12) related analogues. - Further work on the additional portions of the culture extract led to
The crystal structures of the known compounds 9 and 10 are reported for the first time. – This sentence is suitable for discussion, not for introduction.
Page 7.
- Materials and Methods 3.1. General experimental procedures
The general experimental procedures and the apparatus used in the current work are same as that described in our previous report [5]. –
It is clear that instrumental methods have very conservative description, but, if possible, give very short description, or underline even minor changes comparing to the work [5].
In 3.2. Plant and fungal materials
The fungus T. virens QA-8 was isolated from the fresh inner root tissue of the Compositae medical plants A. argyi collected at Qichun of Hubei Province in central China in July 2014 and was identified by analysis of its ITS region of the rDNA. - please, describe primers used for PCR and total length of sequenced ITS.
The BLAST search showed the amplified ITS sequence (GenBank accession no. MK224593) has 99% homology with other members of the genus T. virens LW23 (compared with KT803076.1).
The BLAST search showed the amplified ITS sequence (GenBank accession no. MK224593) has 99% homology with representative member of the genus T. virens LW23 (GenBank accession no. KT803076.1). Show range of homology to other species of the genus for comparison.
3.3. Fermentation, extraction and isolation
The fresh mycelia of T. virens QA-8 were grown on PDA medium at 28 °C for 7 day – The culture of T. virens QA-8 was grown on PDA medium at 28 ď‚°C for 7 day…
Author Response
Dear Referee #3,
Thank you very much for your comments and suggestions to our manuscript entitled “Isolation and Characterization of Antibacterial Carotane Sesquiterpenes from Artemisia argyi Associated Endophytic Trichoderma virens QA-8” (ID: antibiotics-1093075). We have read the comments carefully and have made corrections which we hope meet with approval. Revised portions are highlighted in the manuscript, and the replies (in blue) to your comments, item by item, are listed below:
the manuscript is well-written, but still need some corrections. Some are suggested below:
Page 1. Abstract and through the text
the planted medicinal herb – the grown (cultivated) medicinal herbs
Response: Checked and corrected, as suggested.
Introduction
widely cultured as an medicinal herb - widely cultivated as an medicinal herb
Response: Checked and corrected, as suggested.
A.argyi and its endophytic fungus have been the source of a wide range of biologically active natural products. – Please, specify “its endophytic fungus” by name, and give reference, or remove this sentence.
Response: Checked and corrected, as suggested.
Page 2.
Further work on the additional portions of the culture extract leading to the identification of 12 carotane sesquiterpenes (1–12) (Figure 1), with five new (trichocarotins I–M, 1–5) and seven known (6–12) related analogues. - Further work on the additional portions of the culture extract led to
Response: Checked and corrected, as suggested.
The crystal structures of the known compounds 9 and 10 are reported for the first time. – This sentence is suitable for discussion, not for introduction.
Response: Checked and corrected, as suggested.
Page 7.
Materials and Methods 3.1. General experimental procedures
The general experimental procedures and the apparatus used in the current work are same as that described in our previous report [5]. –
It is clear that instrumental methods have very conservative description, but, if possible, give very short description, or underline even minor changes comparing to the work [5].
Response: Checked and corrected, we have given a very short description.
In 3.2. Plant and fungal materials
The fungus T. virens QA-8 was isolated from the fresh inner root tissue of the Compositae medical plants A. argyi collected at Qichun of Hubei Province in central China in July 2014 and was identified by analysis of its ITS region of the rDNA. - please, describe primers used for PCR and total length of sequenced ITS.
Response: Checked and corrected.
The BLAST search showed the amplified ITS sequence (GenBank accession no. MK224593) has 99% homology with other members of the genus T. virens LW23 (compared with KT803076.1).
The BLAST search showed the amplified ITS sequence (GenBank accession no. MK224593) has 99% homology with representative member of the genus T. virens LW23 (GenBank accession no. KT803076.1). Show range of homology to other species of the genus for comparison.
Response: Thanks to the suggestion. We have re-blasted the amplified ITS 1-4 sequence of T. virens QA8 and found that it has 100% homology with other members of the genus T. virens. The range of homology to other species is shown as below:
3.3. Fermentation, extraction and isolation
The fresh mycelia of T. virens QA-8 were grown on PDA medium at 28 °C for 7 day – The culture of T. virens QA-8 was grown on PDA medium at 28 °C for 7 days…
Response: Checked and corrected, as suggested.
We hope that our revision and the responses above are satisfactory, and that the manuscript is suitable for acceptance in Antibiotics as a full paper. However, if there are any other changes that you would like us to consider, please let me know.
With kind regards!
Bin-Gui Wang
please see our responses in the attached file.
Round 2
Reviewer 1 Report
The authors improved the manuscript and made the requested changes.
Author Response
Comments and Suggestions for Authors
The authors improved the manuscript and made the requested changes.
Response: Thanks to the referee for the comment and suggestion.
Reviewer 2 Report
The authors have completed some edits but the introduction does not follow convention and must be improved. Currently, there is a reference to their new compounds, figure 1, and some of the second paragraphs is describing current methods. Currently, the introduction reads as though it is an abstract. This must be improved and be relevant back ground information, not current study information.
Consider something like the following:
Introduction to the plant and fungus, talk about the general class of compounds found and what they do naturally, discuss the medical significance (health related and antimicrobial activities) as it relates to humans, and then the goal of this project at the end of the introduction.
Minor issues:
Abstract:
Structure-activity relationships (SARs) - the abbreviation is not needed as it is redundant and not used else where in the abstract
Introduction:
line 2: and haves been... Consider revising to: and has been...
line 4 and 25: and so on... The readers do not know what and so on is.. The authors need to list them out or just state a few example.
line 5-6: endophytic fungus (Trichoderma .....): consider revising as the authors list two fungi: endophytic fungi (Trichoderma....).
Author Response
Comments and Suggestions for Authors
The authors have completed some edits but the introduction does not follow convention and must be improved. Currently, there is a reference to their new compounds, figure 1, and some of the second paragraphs is describing current methods. Currently, the introduction reads as though it is an abstract. This must be improved and be relevant back ground information, not current study information.
Consider something like the following:
Introduction to the plant and fungus, talk about the general class of compounds found and what they do naturally, discuss the medical significance (health related and antimicrobial activities) as it relates to humans, and then the goal of this project at the end of the introduction.
Response: Thanks to the suggestion. The Introduction section has been modified, as suggested.
Minor issues:
Abstract:
Structure-activity relationships (SARs) - the abbreviation is not needed as it is redundant and not used else where in the abstract
Response: Corrected, as suggested.
Introduction:
line 2: and haves been... Consider revising to: and has been...
Response: Corrected, as suggested.
line 4 and 25: and so on... The readers do not know what and so on is.. The authors need to list them out or just state a few example.
Response: Corrected, as suggested.
line 5-6: endophytic fungus (Trichoderma .....): consider revising as the authors list two fungi: endophytic fungi (Trichoderma....).
Response: Corrected, as suggested.